# Optimization of Ultrasonic—Assisted Extraction (UAE) Method Using Natural Deep Eutectic Solvent (NADES) to Increase Curcuminoid Yield from *Curcuma longa* L., *Curcuma xanthorrhiza*, and *Curcuma mangga* Val.

**DOI:** 10.3390/molecules27186080

**Published:** 2022-09-18

**Authors:** Desy Rosarina, Dimas Rafi Narawangsa, Nabila Shaffa Rizky Chandra, Eka Sari, Heri Hermansyah

**Affiliations:** 1Department of Chemical Engineering, Faculty of Engineering, Universitas Indonesia, Depok 16424, Indonesia; 2Bioengineering & Biomedical Engineering, Research Centre CoE, Engineering Faculty, Sultan Ageng Tirtayasa University, Cilegon 42434, Indonesia; 3Chemical Engineering, Engineering Faculty, Sultan Ageng Tirtayasa University, Cilegon 42434, Indonesia

**Keywords:** curcumin, turmeric, extraction, ultrasonic-assisted extraction, natural deep eutectic solvents

## Abstract

This study aims to optimize ultrasonic-assisted natural deep eutectic solvents (NADES) based extraction from *C. longa*. Choline chloride-lactic acid (CCLA-H_2_O = 1:1, b/v) was used to investigate the impact of various process parameters such as solvent’s water content, solid loading, temperature, and extraction time. The optimal yield of 79.635 mg/g of *C. longa* was achieved from extraction in 20% water content NADES with a 4% solid loading in 35 °C temperature for 1 h. Peleg’s model was used to describe the kinetics of the optimized ultrasonic-assisted extraction (UAE) method, and the results were found to be compatible with experimental data. The optimum conditions obtained from *C. longa* extraction were then used for the extraction of *C. xanthorriza* and *C. mangga*, which give yields of 2.056 and 31.322 mg/g, respectively. Furthermore, n-hexane was utilized as an anti-solvent in the separation process of curcuminoids extract from *C. longa*, *C. xanthorriza*, and *C. mangga*, which gave curcuminoid recovery of 39%, 0.74%, and 27%, respectively. Solidification of curcuminoids was also carried out using the crystallization method with n-hexane and isopropanol. However, the solution of CCLA and curcuminoids formed a homogeneous mixture with isopropanol. Hence, the curcuminoids could not be solidified due to the presence of NADES in the extract solution.

## 1. Introduction

Curcuminoids are hydrophobic polyphenols found in rhizomes of plants from the *Zingiberaceae* family that are widely cultivated in Asia, including India and Indonesia. Curcuminoids are the main pigment of the plants, giving their rhizomes a bright, yellow-orangish color. These compounds consist of three forms of polyphenols, namely, curcumin (C_21_H_20_O_6_), demethoxycurcumin (C_20_H_18_O_5_), and bisdemethoxycurcumin (C_19_H_16_O_4_). The highest curcuminoid content was found in the rhizomes of the *C. longa* (3–8%), *C. xanthorriza* (2%), and *C. mangga* (3%) [1,2,3].

Curcuminoids are known for having many medical benefits. The pharmacological effects of curcuminoids, such as antiinflammation, antioxidant, antibacterial, anticancer, and antiallergy properties, have been medically proven, both in vitro and in vivo [4]. Curcumin is also known to inhibit angiogenesis and tumor development and to induce apoptosis [5,6]. In addition, based on in silico studies, curcumin is also known to have high affinity for the spike (s) glycoprotein of SARS-CoV-2 and ACE2 which indicates its potential antiviral activity in preventing the transmission of COVID-19 [7].

According to the Joint FAO/WHO Expert Committee, naturally extracted curcuminoids are preferred over chemically synthesized curcuminoids as a food additive. This has led to numerous studies on the most suitable extraction methods for this compound [8,9,10,11,12]. Some conventional methods such as maceration, Soxhlet, and reflux can be used to extract curcumin successfully. However, those methods have some major disadvantages: long extraction time, the use of large amounts of organic solvents, and high temperature that can cause the degradation of curcuminoids, leading to poor efficiency of extraction and high energy consumption.

In recent years, modern methods such as ultrasonic-assisted extraction (UAE) have been researched for greener and more cost-effective extraction of natural resources [10,12]. This method utilizes ultrasonic waves ranging from 20 kHz to 2000 kHz to generate shock waves and various physical phenomena causing the cavitation, i.e., the formation, growth, and explosion, of the cavitation bubbles. The cavitation process results in intense local heating and pressure. Based on the hotspots mechanism, the bubble burst increases at a local temperature of more than 5273 K and a pressure of about 2000 atm [13]. However, this phenomenon does not cause heat damage to the phytochemical because the bubble burst occurs at a fast time with cooling rates in the range of 10^10^ K/s [13]. These extreme conditions break chemical bonds, thus helping the diffusion of solvents to plant cell walls. Therefore, the extraction can be carried out faster even though the process occurs at room temperature [14].

For the extraction of curcuminoids, organic solvents are commonly used. Curcuminoids can easily dissolve in organic solvents such as ethanol, methanol, ethyl acetate, and acetone [8]. However, although the organic solvents have important characteristics for the extraction and the dissolution of natural resources, they have limitations, especially when used in extensive amounts, such as toxicity, non-biodegradability, environmental pollution, lack of selectivity for the desired compound, flammability, and leaving hazardous residue in the extract [15]. Therefore, developing green and sustainable solvents as an alternative to organic solvents is expected to solve these problems. 

NADES (*natural deep eutectic solvent*) is a green solvent composed of natural primary metabolites such as sugars, alcohols, amino acids, amines, vitamins, and carboxylic acids [16]. NADES is formed by mixing quaternary ammonium salts, i.e., hydrogen bond acceptor (HBA), which forms hydrogen bonds with hydrogen bond donors (HBD) [17]. This solvent is called eutectic because a eutectic point will be formed when the two components are mixed in the correct ratio [18]. Compared to organic solvents, NADES has advantages, such as biocompatibility, biodegradability, low toxicity, and low-cost [15]. However, NADES has a high viscosity, which results in its slow diffusivity in the plant matrix during extraction. Therefore, adding water to the mixture of HBA and HBD plays an important role in NADES-based extraction.

In this research, the UAE method is optimized by applying four variations: water content in solvent (NADES), solid loading (*w/v* %), extraction temperature, and time. These variations are carried out based on the one-variable-at-a-time (OVAT) principle, meaning that when the first parameter is varied, the other parameters are set constant. A comparison study has also been carried out between the Soxhlet and UAE-optimized methods. In addition, a kinetics study was conducted using Peleg’s model to evaluate the extraction kinetics parameters and their compatibility with experimental data for UAE extraction of curcuminoids from *C. longa.* The research then continued to optimize the yield of curcuminoid extract from *C. xanthorriza* and *C. mangga* based on the previously optimized parameter obtained. The separation and crystallization methods were also used to produce high purity curcuminoids and maintain the curcuminoid’s physical and chemical stability. Curcuminoids extract is quantified with the absorbance test using UV–VIS spectrophotometry.

## 2. Results and Discussion

### 2.1. Soxhlet Method

The Soxhlet method resulted in 88.476 mg curcuminoid/g from *C. longa* (8.8% *w/w*). The obtained yield was considered as reference, i.e., 100%. Note that the curcuminoid obtained from this experiment is a crude extract, since it is not pure and might still contain some other polyphenols and many other minor compounds. Viscous extract obtained from this process is 1.56 g (31.2% *w/w*) from the original 5 g *C. longa* powder. 

### 2.2. Optimized UAE Method

#### 2.2.1. Water Content (%) Variation

The major drawback of using NADES as an extraction solvent is its high viscosity which can inhibit the diffusion of the solvent to the cell matrix [12]. The dissolution of NADES with distilled water aims to reduce the high viscosity of NADES [15]. Therefore, in this study, the effect of adding water into NADES (20%, 25%, and 30%) was investigated.

The best variation obtained is 20% water content in NADES at 60 min with 43.83 mg/g curcuminoid extract yield. The other two variations, 25% and 30%, gave lower results at 60 min extraction time: 32.58 and 30.74 mg/g, respectively. The complete extraction data will be shown in Figure 1.

As shown in Figure 1, the 20% water content in NADES gave the highest yield of curcuminoids. In ultrasonic extraction, viscosity and surface tension play a significant role [10]. The decrease in viscosity and surface tension causes the formation of cavitation explosions more easily. Fluids with lower viscosity tend to form stable bursts of bubbles and cavitation with a larger magnitude than those with high viscosity [19]. Thus, this helps facilitate easier diffusion of the solvent into the plant sample. However, adding water content of as much as 5% successively into NADES causes a decrease in the yield of curcuminoids. Gabriele (2019), who also investigated the effect of adding water content to NADES, stated that the hydrogen bond between HBA (hydrogen bond acceptor) and HBD (hydrogen bond donor) would weaken with the addition of water content up to 50% [20]. At 75% water content, the bond is completely broken. Therefore, in this experiment, 20% water content in NADES is the optimal condition.

#### 2.2.2. Solid Loading (%) Variation

Mass transfer at the solid–liquid interface is influenced by the proportion of solids and the amount of solvent under specific extraction operating conditions. Therefore, in this study, the effect of % (*w/v*) solid loading (7%, 5%, and 4%) with extraction operating conditions of 20% water content in NADES, temperature 35 °C, particle size 0.25 mm, power 60% (6 s on, 4 s off), for 60 min was studied. 

As shown in Figure 2, the best variation obtained is 4% solid loading at 60 min with 79.67 mg/g curcuminoid extract yield. The other two variations, 5% and 7%, gave lower results at 60 min extraction time: 64.55 and 43.83 mg/g, respectively. 

The yield of curcuminoid extract increased alongside the decrease of solid loading. At a fixed amount of solid matrix, the more solvent volume used, the greater the concentration gradient between the plant cell matrix and the extraction medium. This increases the driving force to release curcuminoids from the sample matrix to the solvent. In addition, a decrease in the solids loading in the solvent also causes cavitation at a lower cavitation threshold [21]. This would significantly increase the extraction rate.

#### 2.2.3. Temperature Variation

Based on Figure 3, the curcuminoids yield obtained increased alongside the increase in temperature from 35 °C to 55 °C. After 60 min of sonication, the maximum curcuminoids yield of 79.64, 80.64, and 84.27 mg/g was achieved from extraction temperatures of 35 °C, 45 °C, and 55 °C, respectively.

The yield of curcuminoid extract increases alongside the increase in temperature. Higher temperature increases the extraction rate because it gives the active compound more kinetic energy. According to diffusivity and solubility of solute, the temperature increase would increase both parameters and thus increase extraction effectivity [22]. Higher temperature would also increase solid material’s pore size, increasing diffusivity further [10]. However, high temperatures can lead to curcumin degradation in the material. This happened at 55 °C temperature, at which the mixture color changed from bright yellow-orangish to dark brown. The dark brown color indicated other compounds, such as ferulic acid, vanillic acid, and vanillin [23,24]. Because of this, 55 °C temperature cannot be used as the optimal condition. Instead, the 35 °C temperature would be chosen since it gave a slightly lower result than the 45 °C and 55 °C, and it used less energy than both temperatures.

#### 2.2.4. Effect of Extraction Time

The extraction process depends on reaction time, and in this study, the extraction process was also observed with variation in extraction time. The longer the extraction was carried out, the more yield was obtained. As shown in Figure 1, Figure 2 and Figure 3, the extraction rate was very fast in the first 20 min. After that, the yields obtained increased gradually up to 1 h. The curcuminoid concentration gradient between the solvent and *C. longa* powder was extensive at the beginning of the extraction process. In addition, the extraction was facilitated by the ease of extraction from the outer matrix during the initial period. As time goes by, the concentration gradient is lower, and the extraction progress is slower because the remaining curcuminoids are in the core of the solid matrix, so this is influenced by diffusion from the solvent. Therefore, there is a limit on the curcuminoids that can be extracted. For every variation, an extraction capacity shows approximately how much curcuminoids can be extracted. The yield will not surpass the extraction capacity, no matter how long the extraction is carried out. The capacity can be calculated empirically using Peleg’s model.

#### 2.2.5. Optimized UAE and Soxhlet Method Comparison

The yield of curcuminoids obtained from the Soxhlet method is considered a value of 100%, which is 88.48 mg/g. In the extraction using the UAE method, a yield of 79.64 mg/g (90%) was obtained with the extraction process only at ±35 °C for 1 h. 

Figure 4 shows how the productivities obtained from extraction with Soxhlet and UAE compare. Although Soxhlet obtained a higher yield, it required a longer time, more solvent, and energy compared to the optimized UA method. Therefore, the productivity of the UAE method (79.64 mg/g·h) is better than that of Soxhlet (7.37 mg/g·h).

The shear force generated due to cavitation along with the shock wave induces physical damage to the plant cell wall, thereby facilitating the release of the compound to be extracted [25]. Diffusion of the solvent also increases with swelling of the plant matrix and hydration by the solvent [26,27]. Ultrasonic wave also increases mass transfer, so solvent diffusion into the plant cell matrix is more effective [10]. Therefore, the UAE only takes a shorter time in the extraction process and requires less energy compared to Soxhlet.

### 2.3. Curcuminoid Extract from C. xanthorriza and C. mangga

The extraction of curcuminoids from *C. xanthorriza* and *C. mangga* with UAE was carried out using the optimized parameters previously obtained from *C. longa* extraction. Based on the experiment, the UAE-optimized method gave yields of 2.06 and 31.32 mg/g for *C. xanthorriza* and *C. mangga*, respectively. The comparison is shown on Figure 5.

The yields obtained from *C. xanthorriza* and *C. mangga* were significantly lower than the one obtained from *C. longa*. *C. xanthorriza* has a curcuminoid content up to 1–2% of its total weight [2], while *C. mangga* has up to 3% [3]. *C. longa* has around 3–8% [1], which makes it the highest source of curcuminoid compared to the other two. *C. xanthorriza* does not contain bisdemethoxycurcumin [28], which is easier to extract compared to curcumin and desmethoxycurcumin [29]. This makes *C. longa*, which contains all three compounds, gave the highest yield. Therefore, *C. longa* is the best option to obtain curcuminoid extract.

### 2.4. Separation and Crystallization Methods

In optimal conditions, separation and crystallization methods were carried out on the curcuminoid extract obtained from *C. longa*, Javanese turmeric, and *C. mangga*. In this study, curcuminoids separated from oleoresin using the anti-solvent precipitation method, where n-hexane was used as antisolvent as curcuminoids are insoluble in it. This process gives curcuminoids recoveries of 39%, 0.74%, and 27% from *C. longa*, *C. xanthorriza*, and *C. mangga*, respectively.

The crystallization process is carried out after the separation process to obtain the curcuminoids in solid form. The mixture of isopropanol-hexane (1:1.5) at 40 °C was used in the solidification process. This process gave curcuminoids recoveries of 22%, 1.29%, and 13% from *C. longa*, *C. xanthorriza*, and *C. mangga*, respectively. However, curcuminoids extract cannot be solidified directly from NADES solution using isopropanol-hexane.

The result showed that both methods can recover some curcuminoids in the extract, which means both methods increase the concentration of curcuminoid extracted. The separation method resulted in a bigger recovery for *C. longa*, but the crystallization method had lower result. The opposite happened to *C. xanthorriza* and *C. mangga*, where the curcuminoid recovery obtained from the crystallization method was bigger than that of the separation method.

The recovery of curcuminoid in the extract can be explained by the antisolvent used in the process. Curcuminoid extract was dissolved in n-hexane which acted as antisolvent. Difference of polarity caused the curcuminoid to precipitate at the bottom of the mixture while being separated from other impurities such as oleoresin [11].

The crystallization process then was attempted to solidify the extract and gain pure curcuminoid powder, but it was unsuccessful. The mixture between NADES and the antisolvent–solvent solution (n-hexane and isopropanol) caused a competition in which both solutions tried to dissolve the curcuminoid. This resulted in precipitation of curcuminoid extract, in which no solids were formed, but some curcuminoids could still be further recovered [11].

### 2.5. Kinetic Model

In this study, the extraction process is modeled with a semi-empirical kinetic model named Peleg’s model. The simplified equation is shown below [12].
(1)$$M_{t}=M_{0}+\frac{t}{K_{1}+K_{2}t}$$
where:M_t_—final concentration of curcumin extract at time t (mg_curcuminoid_/g_material_);M_0_—initial concentration of curcumin extract (mg_curcuminoid_/g_material_);K_1_—Peleg’s rate constant (min g_material_/mg_curcuminoid_);K_2_—Peleg’s capacity constant (g_material_/mg_curcuminoid_);t—extraction time (min).

The equation is made into linear regression equation, with 1/M_t_ as y, 1/t as x, K_1_ as gradient, and K_2_ as intercept. The equation is shown below.
(2)$$\frac{1}{M_{t}}=K_{1}\frac{1}{t}+K_{2}$$

The obtained data would be plotted on a graph to obtain the linear regression equation and coefficient of determination for the models. The model has some kinetic parameters, such as K_1_ (Peleg’s rate constant), K_2_ (Peleg’s capacity constant), B_0_ (initial extraction rate), and C_e_ (curcuminoids concentration at equilibrium) [30].

Peleg’s model was used to describe the extraction kinetic of the optimized UAE method. *C. longa* extraction results are used to make this model. Each variation would have one Peleg’s model to determine their own parameters. The Peleg’s models created from three variations are shown in Figure 6, Figure 7 and Figure 8.

Compatibility of kinetics study with experiment data was determined using RMSE, adjusted R^2^, and E (%) value. Table 1 revealed the results obtained for Peleg’s model.

Table 2 shows the kinetic parameters for Peleg’s model, which include K_1_ (Peleg’s rate constant), K_2_ (Peleg’s capacity constant), B_0_ (initial extraction rate), and C_e_ (curcuminoids concentration at equilibrium).

The Peleg’s model showed a relation between extraction time and the extract concentration in solvent. Figure 6, Figure 7 and Figure 8 showed an increase of curcuminoid extract yield alongside the increase of extraction time. The errors in these models are very miniscule and can be ignored. The model showed a very good coefficient of determination (R^2^), which means a good correlation between each data. The parameters shown can illustrate how well the extraction of each variations went. The bigger the C_e_, more curcuminoid extract can be obtained from the variation that is being carried out. Since this model is linear, it showed that the 55 °C variation has the highest C_e_. However, this is not the case since the 55 °C variation has a risk of curcumin degradation because of high temperature, indicated by color change of the mixture from yellow-orangish to dark brown. While the model showed a good correlation with the experimental data, it cannot show how the extraction went in the experiment. That is why the optimal variation is the 35 °C temperature.

## 3. Materials and Methods

### 3.1. Materials

Fresh *C. longa* was purchased at Tangerang (Banten, Indonesia) local market. Fresh *C. mangga* was commercially purchased from a local business at Garut (West Java, Indonesia). Fresh *C. xanthorriza* was purchased at local market in Jakarta Selatan (DKI Jakarta, Indonesia).

Pure curcuminoids standard was commercially purchased from a retailer at Bantul (Yogyakarta, Indonesia). Choline chloride (95%), lactic acid (90%), analytical grade ethanol (>99%), hexane (>98.5%), and isopropanol (>99.8%) were obtained from Merck (Darmstadt, Germany). A nylon membrane filter with a pore size of 0.45 µm was obtained from Merck Millipore (Carrigtwohill, Ireland). Whatman filter paper (No. 42) with 90 mm pore size was obtained from Whatman (Dassel, Germany).

### 3.2. Preparation of NADES

In this study, NADES was prepared with the heating and stirring method. Choline chloride (HBA) and lactic acid (HBD) were mixed with a 1:2 molar ratio. Water content in NADES was adjusted by adding aquadest. The components were mixed in a beaker glass at 70 °C with constant stirring until clear, homogenous liquid formed. NADES was then stored in a sealed glass at room temperature until use.

### 3.3. Extraction Method 

#### 3.3.1. Soxhlet Extraction

Soxhlet extraction was utilized as the reference method to compare the optimization results performed on the *ultrasound-assisted extraction* (UAE) method. The procedure for this method was carried out based on Patil et al. (2020) [12]. A total of 5 g of *C. longa* powder in the thimble was extracted using ethanol (96%) with a ratio of 1:50 (*w/v*). The extraction process was carried out for 12 h at temperature of ±70 °C. The curcuminoid yield obtained was a reference result (100%).

#### 3.3.2. Ultrasonic-Assisted Extraction

*Ultrasound-Assisted Extraction* was carried out with a NADES-based solvent, choline chloride: lactic acid (1:2), with 250 W rated output power and a frequency of 22 kHz. The extraction was performed with a sonicator probe operated at pulsed mode with a 60% duty cycle (6 s on and 4 s off). The ultrasonic waves were transmitted directly to the mixture for 1 h, with data collection every 10 min. The samples then were diluted, centrifuged, and analyzed quantitatively using UV–VIS spectrophotometry. The effect of water content (%) in NADES, solid loading, extraction temperature, and time were investigated to optimize the process.

### 3.4. Curcumin Quantification

Spectrophotometry UV–VIS was used to analyze the curcuminoids concentration from the samples. The measurements were carried out at a wavelength of 415 nm, giving the absorbance data. The concentrations of curcuminoids (ppm) were calculated using a standard curve of 95% pure curcuminoids and consequently calculated as the yield of curcuminoids (mg/g).

### 3.5. Curcumin Extract Separation

In the present work, curcuminoids and oleoresin were separated using an antisolvent technique with n-hexane. The viscous extract obtained from the optimized UAE method was centrifuged at 6000 rpm for 50 min, and the supernatant was stored in another tube. A known volume of supernatant was mixed with hexane (1:25, *v/v*) and allowed to stand for 24 h. After 24 h, the solution was stirred with a magnetic stirrer at 600 rpm for 3 h. The precipitated mixture was separated from oleoresin by centrifuging the solution at 1000 rpm for 10 min. The recovery of curcuminoids was analyzed quantitatively using spectrophotometry UV–VIS at 415 nm wavelength.

### 3.6. Curcumin Extract Crystallization

The separated curcuminoids from a previous process were further purified employing the crystallization method used by Pawar et al. (2018) [8]. Curcuminoid extract was added to 10 mL of isopropanol and n-hexane mixture (4:6, *v/v*) at 40 °C. The mixture and extract were then cooled in a freezer (temperature below 10 °C) for approximately 1 h. The sediment then was separated from the filtrate using a vacuum filter and then analyzed using UV–VIS spectrophotometry.

## 4. Conclusions

The optimal condition for NADES based UAE in this research is 20% water content, 4% solid loading, 35 °C temperature, and 60 min extraction time. This condition resulted in 79.636 mg curcuminoid extract/g *C. longa*. Peleg’s model was used to describe the kinetics of the optimized UAE method, and the results were found to be compatible with experimental data. Based on the yield obtained, the use of solvents, temperature, and extraction time, the optimized UAE method can be chosen as an alternative to the Soxhlet method. Furthermore, *C. longa* is the best material from which to obtain curcumin extract, since *C. xanthorriza* and *C. mangga* gave a significantly lower result. The separation and crystallization methods are worth carrying out with *C. longa* extract since this extract gave the highest recoveries of curcuminoid, which were 39% from separation and 22% from crystallization. However, curcuminoids could not be solidified due to the presence of NADES in the extract solution.

## Figures and Tables

**Figure 1 molecules-27-06080-f001:**
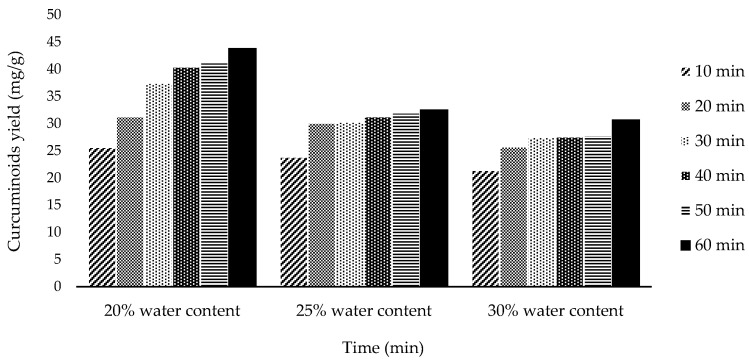
Curcumin extract yield to time graph for water content variation.

**Figure 2 molecules-27-06080-f002:**
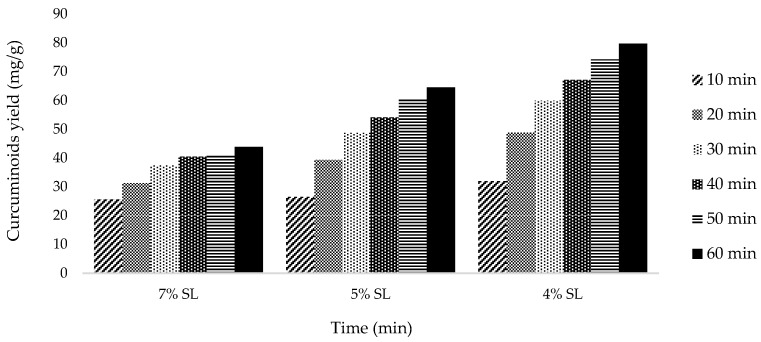
Curcumin extract yield to time graph for solid loading variation.

**Figure 3 molecules-27-06080-f003:**
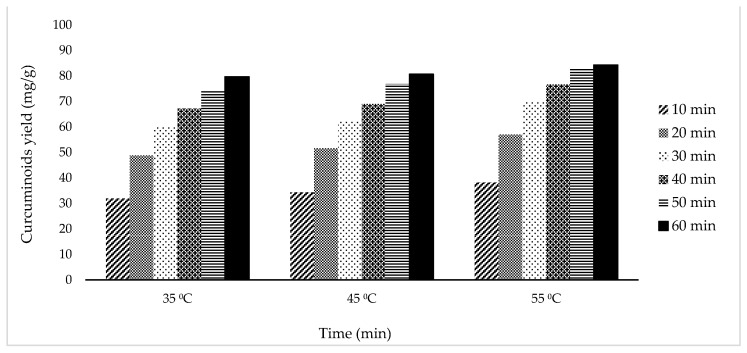
Curcumin extract yield to time graph for temperature variation.

**Figure 4 molecules-27-06080-f004:**
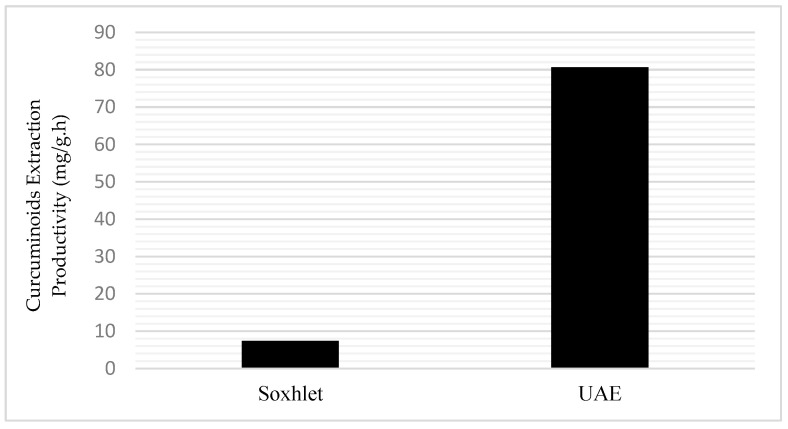
Comparison of extraction productivity obtained from Soxhlet and UAE.

**Figure 5 molecules-27-06080-f005:**
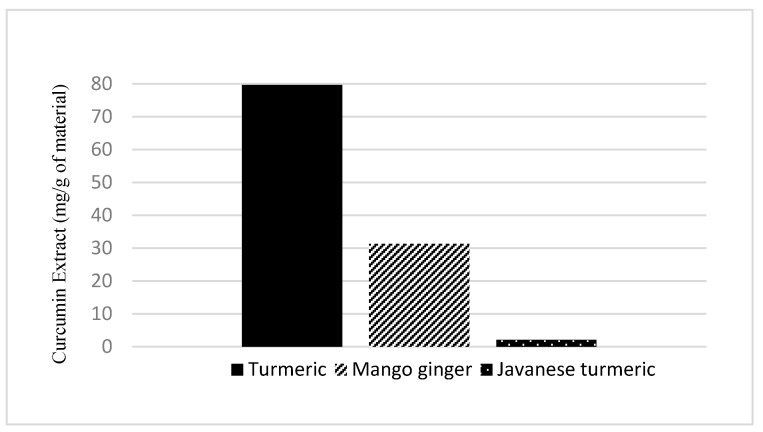
Curcumin extract yield comparison.

**Figure 6 molecules-27-06080-f006:**
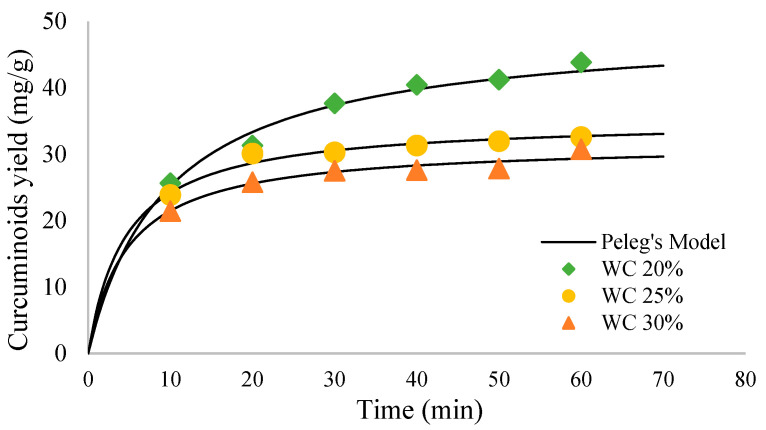
Peleg’s model for water content variation.

**Figure 7 molecules-27-06080-f007:**
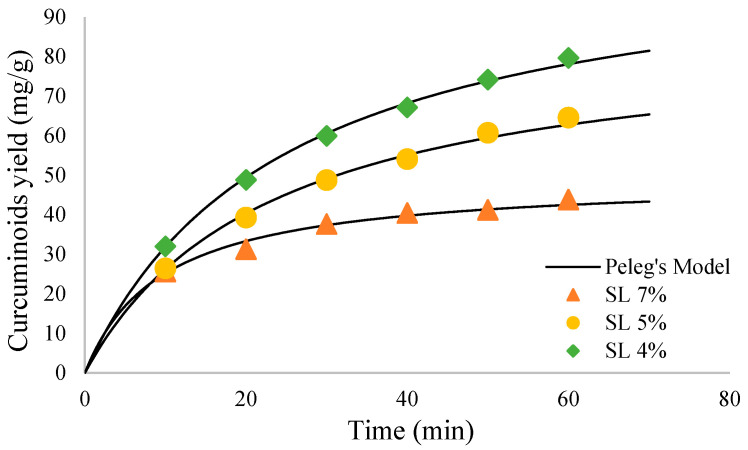
Peleg’s model for solid loading variation.

**Figure 8 molecules-27-06080-f008:**
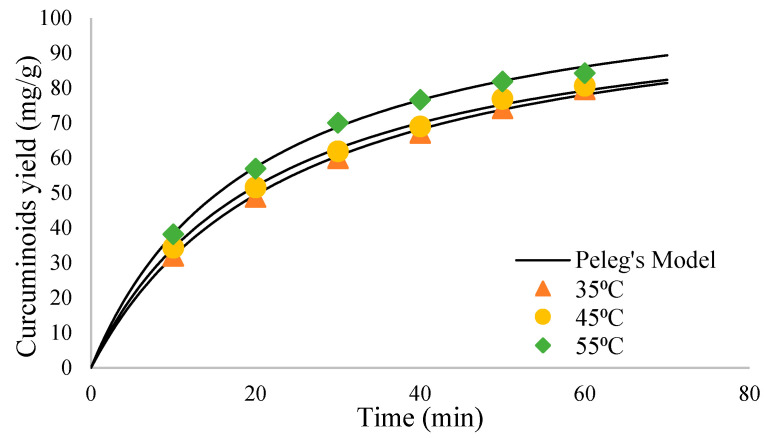
Peleg’s model for temperature variation.

**Table 1 molecules-27-06080-t001:** RMSE, R^2^, and E% value for Peleg’s model.

Variables		RMSE	R^2^	E (%)
Water content (%)	20	0.43	0.97	0.05%
	25	0.26	0.96	0.01%
	30	0.32	0.96	0.01%
Solid loading (%)	7	0.43	0.97	0.05%
	5	0.44	0.99	0.05%
	4	0.35	0.99	0.02%
Extraction temperature (°C)	35	0.35	0.99	0.02%
	45	0.42	0.99	0.04%
	55	0.38	0.99	0.05%
Average		0.38	0.97 *	0.03

* Adjusted R^2^.

**Table 2 molecules-27-06080-t002:** Kinetics parameter for Peleg’s model.

Variables		K_1_ (g min/mg)	K_2_ (g/mg)	B_o_ (mg/g min)	C_e_ (mg/g)
Water content (%)	20	0.19	0.02	5.15	49.26
	25	0.13	0.03	7.70	35.21
	30	0.15	0.03	6.67	31.65
Solid loading (%)	7	0.19	0.02	5.15	49.26
	5	0.27	0.01	3.76	86.96
	4	0.22	0.01	4.50	109.89
Extraction temperature (°C)	35	0.22	0.01	4.50	109.89
	45	0.20	0.01	5.02	107.53
	55	0.18	0.01	5.73	114.94

## Data Availability

Not applicable.

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
