# Peer review of "Optimization of Ultrasonic—Assisted Extraction (UAE) Method Using Natural Deep Eutectic Solvent (NADES) to Increase Curcuminoid Yield from Curcuma longa L., Curcuma xanthorrhiza, and Curcuma mangga Val."

_molecules, 2022, doi:10.3390/molecules27186080_

Round 1
Reviewer 1 Report
I cannot recommend the manuscript Optimization of Ultrasonic-Assisted Extraction (UAE) Method 2 Using Natural Deep Eutectic Solvent (NADES) to Increase Cur-3 cumin Yield from Turmeric (Curcuma longa L.), Javanese Tur-4 meric (Curcuma xanthorrhiza), And Mango Ginger (Curcuma 5 mangga Val.) for publication in the Molecules Journal in its present form and without some major improvements as it contains some poorly explained issues.
*Presenting Results and Discussion as a separate sections makes the manuscript difficult to follow. I believe that standard approach would be more preferable.
*page 6. line 131. The authors need to specify the Figure or Table in which the results are presented.
For example: According to the result presented in Figure 1…The authors should do this also for every subsection
*page 6 line 32. Larger percent of water means less viscosity. “Less viscous solution tends to increase ultrasonic extraction rate…” But the yield decreases with increase the amount of water? The authors should explain this better.
*Why the yields of curcumin from Javanese turmeric and mango ginger are significantly lower i.e. why is harder to extract other forms of curcumin from them?
*Peleg’s model principle (subsection 4.4.) should be introduced before presenting the results within section 2.5.
*The main mechanism of extraction of curcumin using NADES should be better explained in Discussion.
Author Response
Response to Reviewer 1 Comments
Point 1: Presenting Results and Discussion as a separate sections makes the manuscript difficult to follow. I believe that standard approach would be more preferable.
Response 1: The format has been changed according to the suggestions given.
Point 2: page 6. line 131. The authors need to specify the Figure or Table in which the results are presented. For example: According to the result presented in Figure 1…The authors should do this also for every subsection
Response 2: Changes have been made to this point.
Point 3: Page 6 line 32. Larger percent of water means less viscosity. “Less viscous solution tends to increase ultrasonic extraction rate…” But the yield decreases with increase the amount of water? The authors should explain this better.
Response 3: An explanation of point 3 has been presented in section 2.2.1.
Point 4: Why the yields of curcumin from Javanese turmeric and mango ginger are significantly lower i.e. why is harder to extract other forms of curcumin from them?
Response 4: An explanation of point 3 has been presented in section 2.3.
Point 5: Peleg’s model principle (subsection 4.4.) should be introduced before presenting the results within section 2.5.
Response 5: The introduction regarding Peleg’s model has been added.
Point 6: The main mechanism of extraction of curcumin using NADES should be better explained in Discussion.
Response 6: The explanation of NADES mechanisms has been added.
Reviewer 2 Report
In my opinion, the manuscript has some shortcomings in regards to some data analysis and text. Specific comments are given below.
-In general there is no comparsion for curcumin contents in all samples [Turmeric (Curcuma longa L.), Javanese Turmeric (Curcuma xanthorrhiza), And Mango Ginger (Curcuma mangga Val.) ] so please include that in the result section. Also, the results in the results section for which sample of the three selected in the study? "this is not clear".
-The abstract and conclusion need to be more clear.
-why did not use HPLC for curcumin determination for three samples?
Author Response
Response to Reviewer 2 Comments
Point 1: In general there is no comparsion for curcumin contents in all samples [Turmeric (Curcuma longa L.), Javanese Turmeric (Curcuma xanthorrhiza), And Mango Ginger (Curcuma mangga Val.) ] so please include that in the result section. Also, the results in the results section for which sample of the three selected in the study? "this is not clear".
Response 1: The comparison of curcuminoids content in three samples is provided in section 2.3. C. longa was chosen as the best source to extract curcuminoids compared to the other two samples, as stated in section 2.3.
Point 2: The abstract and conclusion need to be more clear.
Response 2: The abstract and conclusion has been revised.
Point 3: why did not use HPLC for curcumin determination for three samples?
Response 3: In this study, we chose spectrophotometry UV-Vis for the quantitative analysis because we needed to find the curcuminoids concentration from 3 variables in 10 min intervals for 1 hour. Therefore, from there, we can choose the optimum parameters.
Round 2
Reviewer 1 Report
The revised manuscript entitled “Optimization of Ultrasonic-Assisted Extraction (UAE) Method Using Natural Deep Eutectic Solvent (NADES) to Increase Curcumin Yield from Turmeric (Curcuma longa L.), Javanese Turmeric (Curcuma xanthorrhiza), And Mango Ginger (Curcuma mangga Val.)” is improved in comparison to the first version, the authors properly responded to questions and remarks, therefore I recommend the manuscript for publishing in the Molecules journal.
Reviewer 2 Report
The authors answered the questions from reviewers, thus, in my opinion, the manuscript is ready to be publishedز